# Molecular and Histopathological Characterization of Metastatic Cutaneous Squamous Cell Carcinomas: A Case–Control Study

**DOI:** 10.3390/cancers16122233

**Published:** 2024-06-15

**Authors:** Alessia Paganelli, Marco Zaffonato, Benedetta Donati, Federica Torricelli, Veronica Manicardi, Michela Lai, Marco Spadafora, Simonetta Piana, Alessia Ciarrocchi, Caterina Longo

**Affiliations:** 1Dermatology Unit, Arcispedale S. Maria Nuova, Azienda Unità Sanitaria Locale IRCCS of Reggio Emilia, 42123 Reggio Emilia, Italy; alessia.paganelli@gmail.com; 2Department of Dermatology, University of Modena and Reggio Emilia, 41121 Modena, Italy; zaffonato.marco@gmail.com; 3Laboratory of Translational Research, Azienda Unità Sanitaria Locale-IRCCS di Reggio Emilia, 42123 Reggio Emilia, Italy; benedetta.donati@ausl.re.it (B.D.); federica.torricelli@ausl.re.it (F.T.); veronica.manicardi@ausl.re.it (V.M.); alessia.ciarrocchi@ausl.re.it (A.C.); 4Clinical and Experimental Medicine PhD Program, University of Modena and Reggio Emilia, 41121 Modena, Italy; michelalai.md@gmail.com (M.L.); spadafora.marco91@gmail.com (M.S.); 5Skin Cancer Center, Azienda Unità Sanitaria Locale—IRCCS di Reggio Emilia, 42123 Reggio Emilia, Italy; 6Pathology Unit, Arcispedale Santa Maria Nuova, Azienda Unità Sanitaria Locale—IRCCS di Reggio Emilia, 40123 Reggio Emilia, Italy; simonetta.piana@ausl.re.it

**Keywords:** gene expression profile, squamous cell carcinoma, risk stratification, metastasis, SCC, personalized medicine

## Abstract

**Simple Summary:**

This study investigates the characteristics of metastatic cutaneous squamous cell carcinoma (cSCC) to improve patient risk stratification. By comparing patients with metastatic and non-metastatic cSCC, we analyzed cSCC skin samples for histological parameters and gene expression profiles. Out of 770 genes tested, 67 were differentially expressed in metastatic cSCC. These were mainly related to immune regulation, skin integrity, angiogenesis, cell migration, and proliferation. The findings suggest that combining histological and molecular profiles can help identify features specific to metastatic cSCC, potentially enhancing patient risk assessment.

**Abstract:**

Background: A subset of patients affected by cutaneous squamous cell carcinoma (cSCC) can exhibit locally invasive or metastatic tumors. Different staging classification systems are currently in use for cSCC. However, precise patient risk stratification has yet to be reached in clinical practice. The study aims to identify specific histological and molecular parameters characterizing metastatic cSCC. Methods: Patients affected by metastatic and non-metastatic cSCC (controls) were included in the present study and matched for clinical and histological characteristics. Skin samples from primary tumors were revised for several histological parameters and also underwent gene expression profiling with a commercially available panel testing 770 different genes. Results: In total, 48 subjects were enrolled in the study (24 cases, 24 controls); 67 genes were found to be differentially expressed between metastatic and non-metastatic cSCC. Most such genes were involved in immune regulation, skin integrity, angiogenesis, cell migration and proliferation. Conclusion: The combination of histological and molecular profiles of cSCCs allows the identification of features specific to metastatic cSCC, with potential implications for more precise patient risk stratification.

## 1. Introduction

Cutaneous squamous cell carcinoma (cSCC) is the second most common non-melanoma skin cancer after basal cell carcinoma [1,2]. Although the majority of cSCCs are promptly treated by surgical excision and have a good prognosis, a quote of cSCCs is associated with a higher likelihood of local invasion, nodal spread (around 4%), recurrence, distant metastasis, and death [3]. Despite the fact that the SCC-related mortality rate is estimated to be around 2%, its high frequency gives reason for the great impact on overall mortality [3,4].

The most well-established risk factors for cSCC include sun exposure, age, fair skin, previous history of actinic keratosis (AK), male sex, and immunosuppression [5,6]. In particular, solid organ transplant recipients seem to be particularly at risk of developing cSCC [7,8]. Depending on the extent of the disease, cSCC is classified into primary and advanced cSCC [9,10,11,12]. A large variety of tumor-intrinsic and tumor-extrinsic factors contribute to the definition of high-risk cSCCs, including, among other factors, tumor diameter and thickness, differentiation, anatomical location, and immune suppression [13]. However, high-risk primary SCCs—as defined by different guidelines [9,10,11,12]—are amenable to curative surgery or radiotherapy [14]. On the contrary, advanced SCCs (either locally advanced or metastatic) cannot be radically treated with surgery and/or radiation therapy, but require systemic treatment [15].

Compared to other forms of cutaneous neoplasms, cSCCs display a very high mutational burden [3,16]. This is at least partially explained by the role of UV radiation in carcinogenesis initiation and promotion. In fact, typical UV-signature mutations have been identified by sequencing of cSCCs [17]. More specifically, *TP53* mutation seems to occur early in cutaneous carcinogenesis and contributes to genomic instability [18]. Subsequently, other pathways are affected, such as CDKN2A, NOTCH, and RAS [19]. Finally, the accumulation of mutations leads to the uncontrolled activation of the NF-kB, MAPK, and PI3K/AKT/mTOR pathways, which mediate epidermal growth factor receptor (EGFR) overexpression and uncontrolled cell proliferation [19,20]. Epigenetic changes may occur as well [19,20]. However, complete understanding of molecular changes in cSCC has not been achieved yet [21].

The aim of the present work was to analyze the features possibly associated with metastatic behavior in cSCC. More specifically, the main endpoint of this case–control study was to compare advanced tumors versus non-metastatic ones (control group, CT) in terms of gene expression profiles as well as clinical and pathological features.

## 2. Materials and Methods

### 2.1. Patient Selection

Patients diagnosed with cSCC between 2007 and 2018 were retrospectively enrolled at the Skin Cancer Center of Arcispedale S. Maria Nuova-IRCCS of Reggio Emilia (CE-875/2020). Other inclusion criteria were the availability of Formalin-Fixed Paraffin-Embedded (FFPE) tumor specimens and a follow-up of at least 1 year from primary tumor excision (in case of living patients). Cases were defined by the presence of histopathologically confirmed loco-regional (lymph-SCC) or distant (met-SCC) metastases.

Patients with and without metastases were matched according to the following criteria: age (±5 years), lesion body site location, and T stage according to the TNM system (AJCC/UICC 8th ed.) [10].

### 2.2. Clinical and Histopathological Data Collection

For each enrolled subject, the following demographic and clinical data were collected: patient age and sex, anatomical location of the primary cSCC, tumor diameter, occurrence of death and specific cause of death, disease stage. Histopathological variables included Breslow thickness, Clark level, differentiation, elastosis, desmoplasia, lympho-vascular or perineural invasion, bone erosion, and tumor margin involvement.

Differentiation grade was evaluated according to WHO classification of skin tumors, 5th edition, based on keratinocyte atypical characteristics such as absence of maturation, mitoses, and dyskeratotic cells [22]. Elastosis was scored according to the criteria proposed by Drexler et al. [23]. Figure 1 shows examples of desmoplastic tumors, as well as different grades of elastosis and differentiation.

### 2.3. Nanostring Analysis

RNA extracted from FFPE samples underwent direct digital RNA counts by nCounter Analysis Systems (Nanostring Technologies, Seattle, WA, USA), with the commercial PanCancer Progression Panel (NanoString Technologies), testing 770 different genes. All the samples were processed according to manufacturers’ instructions at the Laboratory of Translational Research of Azienda USL-IRCCS of Reggio Emilia.

Five slides of 5 μm FFPE tissue were retrieved at the Pathology Unit of Azienda USL-IRCCS of Reggio Emilia for each enrolled patient. Total RNA was extracted by Maxwell^®^ RSC RNA FFPE kit (Promega, Madison, WI, USA); quantity and quality were assessed by NanoDrop 2000 (Thermo Fisher Scientific, Waltham, MA, USA). All RNA samples were suitable for Nanostring analysis (A260/A280 ≥ 1.7 and A260/A230 ≥ 1.8). Detected gene counts were analyzed by nSolver Analysis Software 4.0 (NanoString Technologies). Log2-transformed raw gene counts were normalized on technical controls and housekeeping genes (CNOT4, FCF1, ZC3H14) and finally underwent differential gene expression analysis. Gene expression data are available at the Gene Expression Omnibus (GEO) repository (accession number: GSE253980).

### 2.4. Network Analysis

Bioinformatic analyses on gene expression profiles were performed by R Software v4.0.3 using the following R packages: ggplot2, ggbiplot (function prcomp). Protein interaction networks and Gene Ontology were further evaluated by STRING (https://string-db.org/, (accessed on 1 January 2022).

### 2.5. Statistical Analysis

A one-sided Student *t* test was used to detect differences in gene expression levels, with a *p*-value < 0.05 being considered statistically significant.

Statistical analyses on clinical and pathological characteristics were performed, applying Fisher’s exact test and Kruskal–Wallis test, respectively, for categorical and continuous variables. Survival analyses were performed by R package “Survminer”. Overall survival and disease-associated survival were calculated as the number of months from diagnosis to death due to any cause or associated to the tumor.

## 3. Results

### 3.1. Patient and Tumor Features

A total of 48 patients affected by cSCC were enrolled in the present study, including 24 metastatic and 24 non-metastatic cases. Mean follow-up duration was 3.3 years (range: 2 months–9.5 years). Of the metastatic cases, six had both nodal and distant metastases (met-SCC), while the others had nodal involvement only (lymph-SCC). Clinical and demographic data of the enrolled subjects are listed in Table 1. Of note is that, despite precise matching for sex not being used as a selection criterion, no significant differences were detected in terms of sex distribution among our patient cohorts.

Breslow thickness was slightly higher for metastatic cSCCs (9 ± 4.3 mm vs. 6.3 ± 4.5 mm, *p* = 0.058). No statistically significant differences were detected in terms of Clark levels. Metastatic tumors were more often poorly differentiated (75%) or moderately differentiated (25%). In contrast, nearly half of the cSCCs in the control cohort were well differentiated (46%, *p* = 0.001). The three groups did not differ significantly in terms of grades of elastosis and desmoplasia. No statistically significant differences were detected in the occurrence of lympho-vascular and/or perineural invasion as well as bone erosion, due to the lack of cases presenting with these features in the selected cohort. Incomplete primary tumor excision—with involvement of at least one of the tumor margins—occurred in seven patients with metastatic SCC and in three subjects of the control group, despite such difference not being statistically significant according to our analysis.

With regards to mortality, Kaplan–Meier survival curves underscored significant differences between the three groups both in terms of overall survival (OS) and disease-specific survival (see Figure 2). However, lymph-SCC and met-SCC displayed overlapping patterns in terms of OS.

### 3.2. Gene Expression Profile Associated with Metastatic Behavior of cSCC

All 48 cSCC specimens provided RNA samples suitable for gene expression analysis by digital profiling. Differential analysis of the gene expression profiles between met-SCC and CT resulted in 67 differentially expressed genes (DEGs) (*p* < 0.05). Of these, 40 were downregulated and 27 were upregulated in met-SCC (see Table 2 and Figure 3).

Principal Component Analysis (PCA) based on DEGs showed that these genes were able to segregate samples according to the presence of metastases (Figure 4A).

Samples from the lymph-SCC group tended to be distributed in the middle of the plot in between CT and met-SCC samples, suggesting an increasing trend in gene deregulation together with tumor aggressiveness. Consistently, 52% of the upregulated genes were more expressed even in met-SCC vs. lymph-SCC; on the contrary, 48% of the downregulated gene turned out to be significantly less expressed in lymph-SCC vs. CT (Figure 4B).

Network analysis showed important interaction between DEG gene products, with cell migration being strictly related to angiogenetic pathways, while regulation of the immune response was possibly more closely associated to skin barrier integrity (Figure 5A and Figure 6A). In particular, gene ontology analysis highlighted that downregulated genes in the metastatic cSCC group included genes related to tight junctions, such as CGN (Cingulin), OCLN (Occludin), TJP3 (Tight Junction Protein 3), and CLDN4 (Claudin 4) (Figure 5A,B). Also, genes involved in regulation of the immune response showed lower expression levels in met-SCC compared to CT. Among these were TPSD1 (Tryptase Delta 1) and TPSB2 (Beta 2), IL18 (Interleukin 18), TGFB2R (Transforming Growth Factor beta type-2 Receptor), SPINK5 (Serine Peptidase Inhibitor Kazal Type 5), and PYCARD, which codifies for an inflammasome component (Figure 5A–C). Not surprisingly, genes involved in angiogenesis were upregulated in the metastatic group, especially in the presence of distant—rather than nodal—metastases. These included, among others, VEGFC (Vascular Endothelial Growth Factor C) and CSPG4 (Chondroitin Sulfate Proteoglycan 4), VAV2 (Vav Guanine Nucleotide Exchange Factor 2), and PTGS2 (Prostaglandin-Endoperoxide Synthase 2) (Figure 6A,B). In addition, several genes involved in the regulation of cell proliferation and migration were overexpressed in met-SCCs, such as NR4A3 (Nuclear Receptor Subfamily 4 Group A Member 3), CDK14 (Cyclin Dependent Kinase 14), IL11, PTK2 (Protein Tyrosine Kinase 2), HAS1 (Hyaluronan Synthase 1), as well as PLAU (Plasminogen Activator) (Figure 6A–C).

## 4. Discussion

We performed a pilot study aimed at identifying genes and histopathological characteristics specific to metastatic SCCs.

Different classification systems are currently available for cSSCs, such as AJCC, UICC, BWH (Brigham and Women’s Hospital), NCCN (U.S. National Comprehensive Cancer Network), and EADO (European Association of Dermato-Oncology) [9,10,12,14]. However, a universally accepted staging system for risk stratification in cSCC is still lacking. As a consequence, the definition of “high risk” and “advanced” tumors can vary by the type of classification used, making it difficult to rely on such systems for adequate patient management [24,25].

In a milestone paper by Tokez et al., the authors refined the most important clinical and histopathological variables related to metastatic biological behavior in cSCCs based on two nested case–control studies [26]. These included tumor diameter and thickness, poor differentiation, involvement of the subcutis, perineural or angiolymphatic invasion, male sex, and location on the face [26,27]. A recent metanalysis by Zakhem et al. identified perineural invasion to be the best predictor of metastatic risk, while neoplastic invasion of subcutis appeared as a reliable indicator both for local recurrence and disease-specific death [28].

In our cohort of patients, on the contrary, differentiation grade seemed to be the major determinant for metastatic risk assessment, despite the fact that tumor thickness was also shown to possibly play a role. Despite the relatively low number of cases considered in our analysis, the strength of our study resides in the extremely precise matching of cases and controls, not only in terms of clinical baseline characteristics, but also regarding specific histopathological features (see Section 2.1 Patient Selection). While recent studies suggest lympho-vascular invasion as an important predictor of local and distant recurrence in cSCCs, we did not observe a significant impact of such a feature on patient risk stratification [29]. Similarly, incomplete tumor excision was not found to be specifically associated with a metastatic phenotype [30]. However, due to intrinsic limitations of the study design (such as the small number of enrolled subjects, and the failure to match adequately in terms of tumor thickness and/or differentiation), we would advise not to ignore such parameters and related risk factors when dealing with patients affected by cSCCs in clinical practice. Due to the retrospective nature of patient enrollment, clinical data on immune suppression were not fully available for our cohort, with a subsequent lack of information on such a renowned risk factor [31,32]. Moreover, a possible bias in the interpretation of our data resides in the lack of information regarding PD-L1/PD-1 expression in the two study groups, which could potentially bring new insights in terms of clinical and therapeutic implications [33].

The collection of clinical data also allowed us to draw few conclusions on patient survival in relation to the absence or presence of metastases. As for the leading cause of death, in fact, patients with metastatic cSCCs were more likely to die from their cutaneous malignancies. While tumor-related mortality accounted for 41.6% of deceased patients in the metastatic group, none of the subjects included in the CT cohort died from tumor-related causes (see Table 1). The relatively high number of deaths for other causes in such subgroup is possibly explained by the old age of our sample and the long follow-up.

Interest is growing toward more precise methods for metastasis risk patient stratification in cSCC [34]. A 40-gene expression profile test has already been validated to stratify patients affected by cSCCs into three main risk categories (low, moderate, and high risk), with significant differences in terms of the occurrence of metastases, varying from 6% in low-risk subjects up to above 50% in high-risk cases [35].

However, other potentially useful candidate genes for broader assays emerge from our data. Not surprisingly, genes involved in tight junction formation were found to be downregulated in metastatic cSCCs; loss of intercellular adhesion is, in fact, crucial for cell migration, angiolymphatic invasion, and eventually metastasis. Tight junction impairment had already been described in the setting of cSCC, especially in association with the invasion of adjacent tissues or metastatic behavior [36,37].

The antitumoral immune response is of particular interest in the setting of dermato-oncology, and there is evidence that metastatic spread is related to a specific phenotype of the lymphocytic infiltrate in skin neoplasms [38]. However, the occurrence of cutaneous cancers is also notably associated with UV-mediated chronic inflammation, which underscores the complexity of the immune cell interplay in skin oncogenesis [39]. We found several genes encoding for proteins involved in immune regulation to be downregulated in the met-SCC group. Among these, for example, *TPSD1* and *TPSB2* are markers of mast cell activation. However, the role of mast cells in the setting of human cancers is controversial; while tumor-associated mast cells have been associated with a better prognosis in the setting of ovarian cancer, B-cell lymphoma, and esophageal adenocarcinoma, poorer outcomes have been highlighted for melanoma, lung or breast cancer [40]. A recent study showed that patients affected by oral SCC with a low mast cell density in tumor-associated stroma had significantly lower overall survival [41]. Reduced expression was also found for *IL18*, *TGFBR2*, and *PYCARD* in the met-SCC group. IL18 is a cytokine with pleiotropic effects, ranging from the repair of the epithelial barrier up to immune polarization towards a Th1 (T helper 1) and NK (natural killer) phenotype. Such a response is crucial for antitumoral immunity, therefore potentially explaining its reduced expression in metastatic samples [42]. TGFBR2 is implicated in the crosstalk between fibroblasts and cancer cells and its protective role has already been described in mammary cancer [43]. PYCARD expression is closely related to inflammasome activation; however, recent literature demonstrated PYCARD to act as a tumor suppressor in keratinocytes through p53 activation [44]. Another publication highlighted that *PYCARD* methylation in SCC cell lines leads to reduced expression of PYCARD protein and, as a consequence, impaired release of downstream cytokines [45].

The met-SCC group displayed upregulation of genes involved in cell proliferation and migration, including *IL11*, *PTK2*, and *PLAU*. PLAU, through binding of its receptor PLAUR, promotes cell invasions through activation of proteolytic cascade and indirectly enables the release of angiogenetic factors such as VEGFC. A recent publication confirmed the increased activation of the PLAU pathway in metastatic cSCCs and suggested PLAUR to be a new putative therapeutic target in this scenario [46]. *PTK2* encodes for a protein found in the focal adhesions and is therefore defined as a member of the FAK (focal adhesion kinase) family. FAK activation is crucial for cell interactions with the extracellular stroma and cell migration. Some evidence already exists for FAK hyperactivation in the setting of other neoplasms, such as colorectal cancer [47]. Notably, IL11 promotes epithelial-to-mesenchymal transition, and has recently been found to act through the activation of MMP-13 (matrix metalloproteinase 13); the same study described a direct correlation between IL11 expression levels and disease stage in oral SCC [48].

Lastly, hyper-expression of genes involved in neo-angiogenesis emerged from our analysis in the met-SCC group. These included *VEGFC*, *PTGS2*, and *CSPG4*. The *PTGS2* gene encodes for the COX-2 (cyclooxygenase-2) enzyme, which is produced in tumor endothelial cells and is necessary for cancer cell migration. COX-2 in endotheliocytes leads to prostaglandin production, which, in turn, eventually leads to VEGF secretion. The activation of this pathway has already been demonstrated in head and neck SCCs [49]. Interestingly, *CSPG4* has also been found among upregulated genes; *CSPG4* has already been proposed as a putative diagnostic marker in aggressive forms of SCC with an active role in cancer progression [50]. Moreover, specific blocking antibodies against CSPG4 have been described to reduce tumor growth and metastasis in vivo [51].

Despite the promising results of our preliminary study, we acknowledge some major limitations, including the small number of tumors examined, the already mentioned restrictions in histopathological evaluation, and the lack of validation of this gene signature in an independent cohort of tumors. Moreover, we split the metastatic cSCC cohort into two subgroups (met-SCC and lymph-SCC), which led to a sample ratio of six vs. twenty-four in comparing gene expression profiles of met-SCC to the control group. This unequal distribution possibly represents a limitation in the interpretation of the subsequent transcriptomic analyses.

As a consequence, it is currently impossible to draw any conclusions on the immediate clinical implications of our results. Based on our preliminary findings, we plan to perform further studies aimed at validating such histological and molecular features in an independent cohort of subjects. Moreover, larger cohorts are needed to determine whether the differentially expressed genes identified by our study are exhaustive for correct patient prognostic classification.

## 5. Conclusions

Our data highlight the presence of specific histopathological and molecular characteristics in metastatic cSCCs. Most of the identified genes are related to immune regulation, angiogenesis, cell migration, and skin barrier integrity. Based on this pilot study, we aim at identifying a specific genetic signature for metastatic cSCC, after further validation in an independent cohort of subjects.

Gene expression profile panels not only open a new scenario in patient prognostic stratification but also bring to light new potential therapeutic targets for advanced or metastatic forms. Potentially, deeper knowledge of a metastasis-specific signature would allow us early identification of high-risk patients and a more active surveillance dedicated to such a subgroup of subjects.

## Figures and Tables

**Figure 1 cancers-16-02233-f001:**
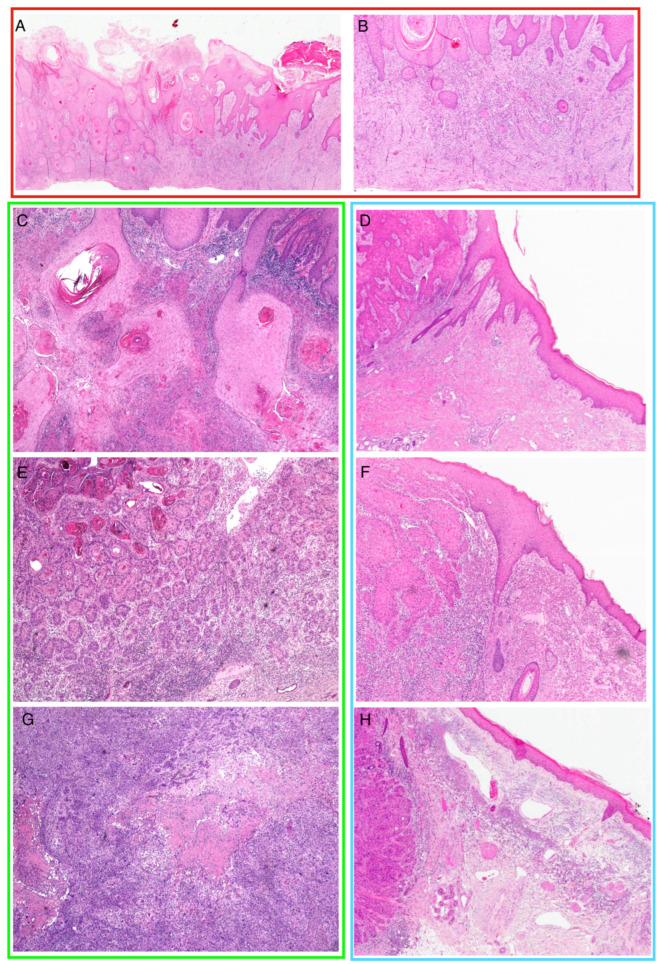
Examples of desmoplasia (red square), elastosis (blue square), and differentiation (green square) in histopathological specimens from our cohort. Panels (**A**,**B**): a case of SCC with a prominent desmoplastic reaction. Solid neoplastic nests infiltrate a fibrotic dermis, with a mild inflammatory infiltrate and many newly formed small vessels (magnification 20× panel (**A**); 40× panel (**B**)). High-magnification (HE 40×) representative examples of different grades of differentiation (panels (**C**,**E**,**G**)) and elastosis (panels (**D**,**F**,**H**)). Panel (**C**): representative example of well-differentiated SCC. Solid neoplastic keratinizing nests infiltrate the dermis with pushing borders. Panel (**E**): moderately differentiated SCC; keratinizing foci alternate with neoplastic nests and cords exhibiting nuclear atypia. Panel (**G**): poorly differentiated cSCC. When poorly differentiated, SCCs grow in solid nests with wide necrotic areas and show striking cytological atypia. Panel (**D**) is an exemplified case of absent or very low score in tumor-associated elastosis grade. At high magnification, the normal eosinophilic collagen fibers are preserved, also near the advancing front of the tumor. Panel (**F**): In intermediate (score 2) tumor-associated elastosis, there is an increase in the superficial basophilic material in the dermis, although the overall architecture, including the vascular network, is maintained. Panel (**H**) is a representative example of score-3 tumor-associated elastosis; a homogeneous basophilic zone is present in the dermis, with complete loss of the normal fiber distribution.

**Figure 2 cancers-16-02233-f002:**
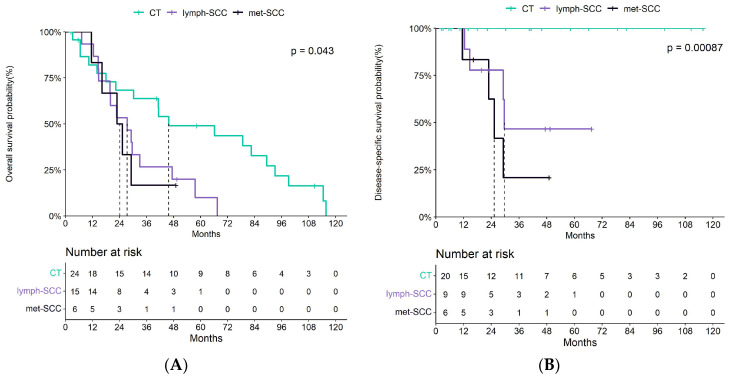
(**A**,**B**) Kaplan–Meier curves showing the overall survival probability in cSCC patients without any metastasis (CT), and with nodal (lymph-SCC) and distant (met-SCC) metastases.

**Figure 3 cancers-16-02233-f003:**
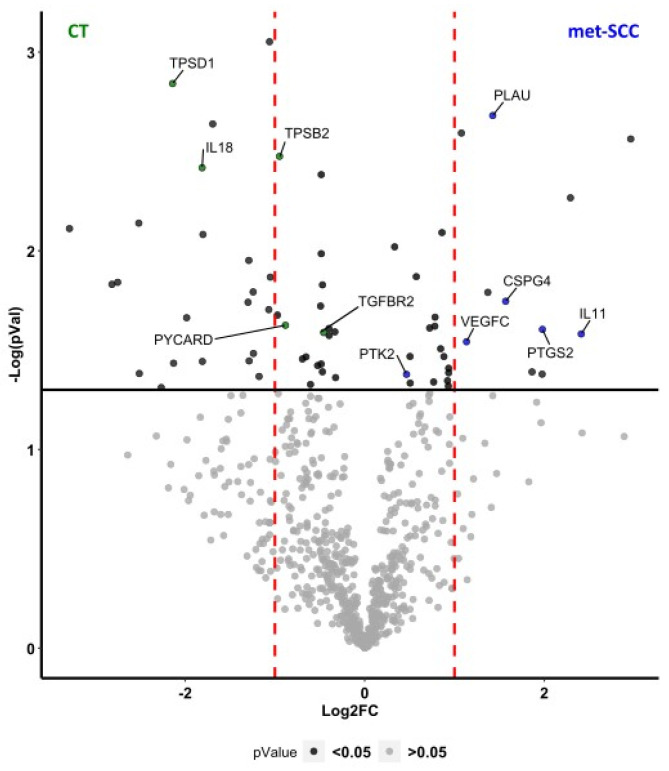
Volcano plot displaying DEGs between metSCC and CT patients. DEGs were considered significant for *p*-value ≤ 0.05 (black dots). Red dashed lines represent absolute FC ˃ 2.

**Figure 4 cancers-16-02233-f004:**
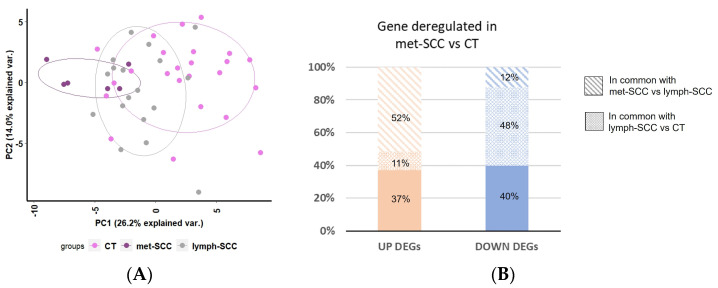
(**A**) Principal Component Analysis (PCA) shows the variance between CT (pink), lymph-SCC (grey), and met-SCC (purple) samples explained by DEGs (*p*-value ≤ 0.05). We created a PCA plot with the function “prcomp” and we represented it using the package “ggbiplot” with R version 4.4.0. Clusters defined by ellipses were obtained by automatic classification of samples based on PCA scores. The center of each ellipse represents the barycenter of the points belonging to the same cluster, while the main axis represents the variance. (**B**) Distribution of DEGs in met-SCC vs. CT considering the percentage of genes in common with the other group’s comparison.

**Figure 5 cancers-16-02233-f005:**
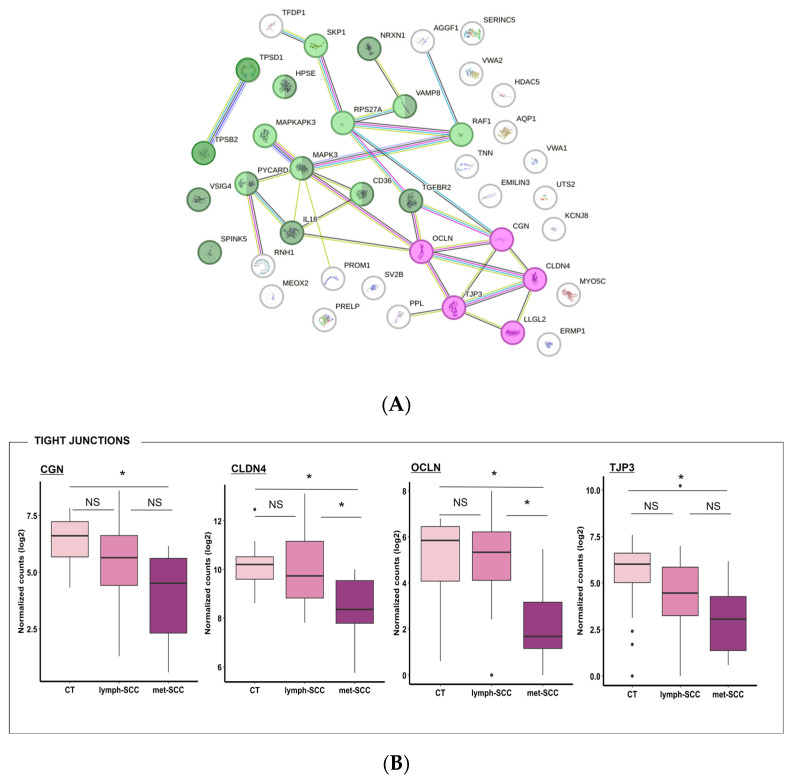
(**A**). Protein–protein interaction network representing genes significantly downregulated (*p* < 0.05) in met-SCC vs. CT samples. These DEGs partake in tight junction organization (purple circles; Hsa04530 *p* = 0.0020) and were related to immune response (green circles; GO 0032101 *p* = 1.26 × 10^−5^, Hsa168249 *p* = 0.007). Network was created using String-DB tool using default setting. (**B**,**C**) Boxplot showing the expression of the main representative downregulated genes belonging to these groups in CT, lymph-SCC, and met-SCC. * *p* < 0.05.

**Figure 6 cancers-16-02233-f006:**
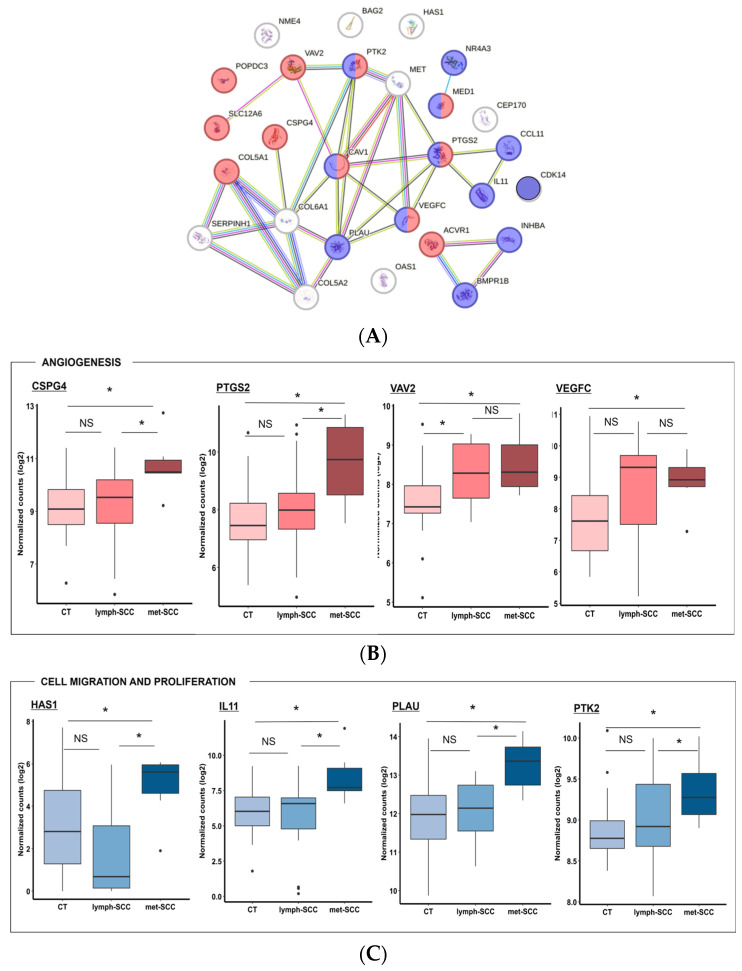
(**A**) Protein–protein interaction network representing genes significantly upregulated (*p* < 0.05) in met-SCC vs. CT samples. These DEGs partake in cell proliferation and migration (blue circles; GO 0030335 *p* = 2.21 × 10^−5^; GO 0030334 *p* = 7.03 × 10^−5^; GO 0042127 *p* = 0.0017) and angiogenesis (red circles; GO 0001525 *p* = 9.59 × 10^−7^; GO 0072359 *p* = 1.26 × 10^−5^). Network was created using String-DB tool using default setting. (**B**,**C**) Boxplots showing the expression of the main representative upregulated genes included in these groups in CT, lymph-SCC, and met-SCC. **p* < 0.05.

**Table 1 cancers-16-02233-t001:** Clinical and pathological features of patients included in this study. Of note is that lymphatic or vascular involvement and perineural invasion are not shown in Table 1 because, in all of the groups, there were no patients with these histopathologic features.

	CT (N = 24)	Lymph-SCC (N = 18)	Met-SCC (N = 6)	Total (N = 48)	*p* Value
**Sex**					0.293
F	8 (33.3%)	5 (27.8%)	0 (0.0%)	13 (27.1%)	
M	16 (66.7%)	13 (72.2%)	6 (100.0%)	35 (72.9%)	
**Age**					0.183
Mean (SD)	81.583 (7.454)	81.389 (7.655)	76.833 (4.956)	80.917 (7.310)	
**Site**					0.980
Lower limbs	2 (8.3%)	2 (11.1%)	0 (0.0%)	4 (8.3%)	
Upper limbs	5 (20.8%)	3 (16.7%)	2 (33.3%)	10 (20.8%)	
Head/neck	14 (58.3%)	10 (55.6%)	4 (66.7%)	28 (58.3%)	
torso	3 (12.5%)	3 (16.7%)	0 (0.0%)	6 (12.5%)	
**Dimension**					0.899
Mean (SD)	2.733 (2.311)	2.917 (2.036)	2.167 (0.489)	2.731 (2.047)	
**Dimension (grouped)**					1.000
>2 cm	11 (45.8%)	9 (50.0%)	3 (50.0%)	23 (47.9%)	
≤2cm	13 (54.2%)	9 (50.0%)	3 (50.0%)	25 (52.1%)	
**Breslow**					0.058
Mean (SD)	6.348 (4.519)	9.500 (4.885)	7.500 (1.761)	7.622 (4.564)	
NA	1	2	0	3	
**Clark**					0.223
II	4 (17.4%)	0 (0.0%)	0 (0.0%)	4 (8.9%)	
III	3 (13.0%)	3 (18.8%)	0 (0.0%)	6 (13.3%)	
IV	9 (39.1%)	3 (18.8%)	3 (50.0%)	15 (33.3%)	
V	7 (30.4%)	10 (62.5%)	3 (50.0%)	20 (44.4%)	
NA	1	2	0	3	
**Differentiation**					0.006
Well/moderately differentiated	17 (70.8%)	5 (27.8%)	1 (16.7%)	23 (47.9%)	
Poorly differentiated	7 (29.2%)	13 (72.2%)	5 (83.3%)	25 (52.1%)	
**Desmoplasia**					
0	17 (70.8%)	12 (70.6%)	4 (66.7%)	33 (70.2%)	0.999
I/II/III	7 (29.2%)	5 (29.4%)	2 (33.3%)	14 (29.8%)	
NA	0	1	0	1	
**Elastosis**					0.702
+	10 (43.5%)	5 (31.2%)	3 (50.0%)	18 (40.0%)	
++/+++	13 (56.5%)	11 (68.8%)	3 (50.0%)	27 (60.0%)	
N-Miss	1	2	0	3	
**Margins**					0.122
Free	21 (87.5%)	13 (76.5%)	3 (50.0%)	37 (78.7%)	
Affected	3 (12.5%)	4 (23.5%)	3 (50.0%)	10 (21.3%)	
NA	0	1	0	1	
**T, N (AJCC/UIC 8th ed)**					0.245
T1N0	12 (50.0%)	7 (41.2%)	2 (33.3%)	21 (44.7%)	
T1N1	0 (0.0%)	1 (5.9%)	1 (16.7%)	2 (4.3%)	
T2N0	9 (37.5%)	4 (23.5%)	3 (50.0%)	16 (34.0%)	
T2N1	0 (0.0%)	2 (11.8%)	0 (0.0%)	2 (4.3%)	
T3N0	3 (12.5%)	1 (5.9%)	0 (0.0%)	4 (8.5%)	
T3N1	0 (0.0%)	2 (11.8%)	0 (0.0%)	2 (4.3%)	
NA	0	1	0	1	
**Stage** **(AJCC/UIC 8th ed)**					<0.001
I	12 (50.0%)	0 (0.0%)	0 (0.0%)	12 (25.5%)	
II	9 (37.5%)	0 (0.0%)	0 (0.0%)	9 (19.1%)	
III	3 (12.5%)	12 (70.6%)	0 (0.0%)	15 (31.9%)	
IV	0 (0.0%)	5 (29.4%)	6 (100.0%)	11 (23.4%)	
NA	0	1	0	1	
**Disease-specific mortality**					<0.001
No	20 (100.0%)	5 (45.5%)	2 (33.3%)	27 (73.0%)	
Yes	0 (0.0%)	6 (54.5%)	4 (66.7%)	10 (27.0%)	
NA	4	7	0	11	
**Overall mortality**					0.449
No	5 (20.8%)	1 (5.9%)	1 (16.7%)	7 (14.9%)	
Yes	19 (79.2%)	16 (94.1%)	5 (83.3%)	40 (85.1%)	
NA	0	1	0	1	

**Table 2 cancers-16-02233-t002:** Genes significantly deregulated in met-SCC vs. CT (*p* < 0.05).

UP Regulated Genes			DOWN Regulated Genes		
Gene	Absolute Fold Change	*p* Value	Gene	Absolute Fold Change	*p* Value
*ACVR1*	1.49	0.013	*AGGF1*	0.71	0.019
*BAG2*	1.82	0.008	*AQP1*	0.62	0.035
*BMPR1B*	3.94	0.042	*CD36*	0.41	0.011
*CAV1*	1.65	0.024	*CGN*	0.18	0.041
*CCL11*	4.90	0.005	*CLDN4*	0.29	0.036
*CDK14*	1.84	0.034	*EMILIN3*	0.41	0.018
*CEP170*	1.42	0.034	*ERMP1*	0.48	0.001
*COL5A1*	1.90	0.045	*HDAC5*	0.72	0.010
*COL5A2*	1.70	0.046	*HPSE*	0.72	0.041
*COL6A1*	1.91	0.041	*IL18*	0.29	0.004
*CSPG4*	2.97	0.018	*KCNJ8*	0.70	0.038
*HAS1*	3.64	0.041	*LLGL2*	0.42	0.033
*IL11*	5.32	0.026	*MAPK3*	0.79	0.026
*INHBA*	2.59	0.016	*MAPKAPK3*	0.48	0.014
*MED1*	1.26	0.010	*MEOX2*	0.51	0.021
*MET*	1.72	0.022	*MYO5C*	0.64	0.034
*NME4*	2.11	0.003	*NRXN1*	0.42	0.016
*NR4A3*	1.91	0.048	*OCLN*	0.14	0.015
*OAS1*	1.80	0.031	*PPL*	0.48	0.020
*PLAU*	2.69	0.002	*PRELP*	0.10	0.008
*POPDC3*	7.80	0.003	*PROM1*	0.25	0.022
*PTGS2*	3.94	0.025	*PYCARD*	0.54	0.024
*PTK2*	1.38	0.042	*RAF1*	0.76	0.025
*SERPINH1*	1.72	0.024	*RNH1*	0.72	0.015
*SLC12A6*	1.42	0.046	*RPS27A*	0.71	0.004
*VAV2*	1.91	0.039	*SERINC5*	0.71	0.037
*VEGFC*	2.19	0.029	*SKP1*	0.80	0.043
			*SPINK5*	0.29	0.008
			*SV2B*	0.23	0.037
			*TFDP1*	0.76	0.027
			*TGFBR2*	0.73	0.026
			*TJP3*	0.21	0.049
			*TNN*	0.31	0.002
			*TPSB2*	0.52	0.003
			*TPSD1*	0.23	0.001
			*UTS2*	0.41	0.036
			*VAMP8*	0.66	0.047
			*VSIG4*	0.44	0.043
			*VWA1*	0.17	0.007
			*VWA2*	0.15	0.014

## Data Availability

The data that support the findings of this study are available from the corresponding author upon reasonable request. Gene expression data are available at the Gene Expression Omnibus (GEO) repository (accession number: GSE253980).

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
