# Peer review of "Molecular and Histopathological Characterization of Metastatic Cutaneous Squamous Cell Carcinomas: A Case–Control Study"

_cancers, 2024, doi:10.3390/cancers16122233_

Round 1

Reviewer 1 Report

Comments and Suggestions for Authors

The authors set out to build a combined pathological and molecular profile to assess the metastatic potential of cutaneous squamous cell carcinoma (cSCC). 48 formalin-fixed paraffin-embedded cSCC samples were selected, with a 1:1 ratio for cases (metastatic cSCC) and controls (non-metastatic cSCC) matched for age, lesion site, and T stage. For each sample clinical and histopathological data were collected as well as performing transcriptomic analyses using the commercially available PanCancer Progression Panel (NanoString Technologies). Degree of differentiation was the most significant clinicopathological factor with tumor thickness coming second to distinguish between metastatic cSCCs and control samples. Following differential gene expression analysis, a gene set of 33 genes was extracted: 11 genes were upregulated in metastatic cSCC playing a role in tight junction formation and regulation of immune response. 22 genes were downregulated in metastatic cSCC, which were associated with regulation of proliferation, migration, and angiogenesis.

The manuscript starts off with a structured introduction stating the problem of missing risk stratification strategies for cSCC leading to the aim of the project. The methodology, results and discussion sections are organized following the same structure, respectively, of (i) clinicopathological data, then turning to (ii) differential gene expression and (iii) gene ontology network analyses.

(i) clinicopathological data:

Patients in the sample and control groups were matched by age, lesion body site and T stage. Reasons for not matching also for sex should be included in the manuscript. In addition, scoring descriptions for “degree of differentiation” as well as for “degree of solar elastosis” should be defined in the methods section. These could also be visualized by a figure presenting histopathological sections for each category.

(ii) differential gene expression:

Following differential gene expression analysis, the metastatic cSCC cohort was split into two subgroups: 6 met-SCC and 16 lymph-SCC. This division, however, leads to a sample ratio of 6 vs 24 in comparing met-SCC to the control group. As a result, this unequal distribution may undermine the subsequent transcriptomic analyses and should be included as limitation in the discussion section.

The authors may consider to perform subsequent analyses also with the lymph-SCC group (compared with met-SCC and the control groups), and / or the initial combined metastatic cSCC cohort of 24 samples.

(iii) gene ontology network analysis:

Gene ontology (GO) can give hints on the biological function of genes and proteins. These pathways can be named in an imprecise manner, such as some stated in the manuscript (“regulation of the immune response” or “regulation of cell proliferation”). In these cases, the effect of the gene set is not clear. In the discussion section, especially for the “regulation of the immune response” it seems as though this GO-term was interpreted as “antitumoral immune response”, while showing at the same time, that several genes herein are not active in “antitumoral immune responses”. All genes are individually discussed – here, it should always be included, in which patient group the respective gene was up- or downregulated.

As the manuscript states, validated risk stratification strategies for cSCC are currently not available. Multiple studies have tried to meet this relevant subject, producing just as many different gene sets with the potential to predict the further course of cSCC disease – yet lacking clinical validation. Here, the authors announce a “molecular and histopathological signature” taking an adequate approach with data collection and transcriptomic analysis.

The manuscript would benefit from distinctively stating the final combined (molecular and histopathological) signature and discussing the advantage of this signature in comparison to the multiple other signatures already published. In this context, it may be advisable to show the prognostic potential of the molecular and histopathological signature on their own and in combination.

Additional comments

In the results section (lines 117-119), supporting notes on how to fill out this section were accidentally included in the manuscript.

Table 1 has a formatting / shifting error in the category “Breslow thickness”. In the following sections the met-SCC cohort is split into two groups – it may be advisable to show the clinical data for both subgroups separately.

Interpretation of the Kaplan-Meier curves (lines 132-138) in the results section belongs in the discussion section.

Please state reasons for showing all 67 significantly deregulated genes in Table 2 and not only those 33 genes chosen according to the combination of p-value and fold change, which are included in the gene signature.

In Figure 2, the reference group should be stated in the figure legend. Relevant genes mentioned in the discussion section may be labelled in the volcano plot.

In Figure 3, labelling of the patient groups should be consistent with the rest of the manuscript. Additionally, the meaning of the ellipses should be explained in the figure legend.

In Figures 4 and 5 all ladder plots should have the same size. Any significances should be added to the plots and stated in the figure legends (also if there are none). For better understanding, the title of the gene sets depicted in the ladder plots can be included in the figure legends.

Figure 5 is mirrored upside down.

For all figures, a higher resolution is necessary.

In the results section (line 101/102), it is stated that the data are available at the GEO repository (release scheduled for Jan 2025). This should be included in the data availability statement.

An ethics statement should be included concerning the informed consent of all patients or an ethical vote, that such a consent is not necessary due to the retrospective nature of the study.

Comments on the Quality of English Language

Minor language errors did not limit understanding of the manuscript.

Author Response

The authors set out to build a combined pathological and molecular profile to assess the metastatic potential of cutaneous squamous cell carcinoma (cSCC). 48 formalin-fixed paraffin-embedded cSCC samples were selected, with a 1:1 ratio for cases (metastatic cSCC) and controls (non-metastatic cSCC) matched for age, lesion site, and T stage. For each sample clinical and histopathological data were collected as well as performing transcriptomic analyses using the commercially available PanCancer Progression Panel (NanoString Technologies). Degree of differentiation was the most significant clinicopathological factor with tumor thickness coming second to distinguish between metastatic cSCCs and control samples. Following differential gene expression analysis, a gene set of 33 genes was extracted: 11 genes were upregulated in metastatic cSCC playing a role in tight junction formation and regulation of immune response. 22 genes were downregulated in metastatic cSCC, which were associated with regulation of proliferation, migration, and angiogenesis. 

The manuscript starts off with a structured introduction stating the problem of missing risk stratification strategies for cSCC leading to the aim of the project. The methodology, results and discussion sections are organized following the same structure, respectively, of (i) clinicopathological data, then turning to (ii) differential gene expression and (iii) gene ontology network analyses. 

Thank you very much for the kind comments. We sincerely appreciated your insights which gave us the chance to modify the manuscript according to the following suggestions, therefore improving its quality and readability.

(i) clinicopathological data: 

Patients in the sample and control groups were matched by age, lesion body site and T stage. Reasons for not matching also for sex should be included in the manuscript. In addition, scoring descriptions for “degree of differentiation” as well as for “degree of solar elastosis” should be defined in the methods section. These could also be visualized by a figure presenting histopathological sections for each category. 

Despite not precise matching for sex due to patient consecutive enrollment, no significant differences were detected in term of sex distribution in our cohort of patients. We also added a supplementary figure with graphical examples of differentiation degrees and elastosis degrees.

(ii) differential gene expression:

Following differential gene expression analysis, the metastatic cSCC cohort was split into two subgroups: 6 met-SCC and 18 lymph-SCC. This division, however, leads to a sample ratio of 6 vs 24 in comparing met-SCC to the control group. As a result, this unequal distribution may undermine the subsequent transcriptomic analyses and should be included as limitation in the discussion section. 

The unequal distribution of the aforementioned groups has been listed as a limitation in the discussion.

The authors may consider to perform subsequent analyses also with the lymph-SCC group (compared with met-SCC and the control groups), and / or the initial combined metastatic cSCC cohort of 24 samples. 

We thank the Reviewer for this interesting comment, indeed the initial aim of our work was to compare patients with a metastatic phenotype with control samples. However, when we focused on the patients with the most aggressive phenotype, who developed both lymph nodal and distant metastases, we identified a gene expression program able to segregate our overall cohort into three groups where patients with lymph nodal metastases were set in the middle (see PCA plot). Thus, we proposed a model in which the presence of lymph nodal metastases can be considered a transition state between the indolent and the most aggressive disease. To strengthen this hypothesis, we provided a further analysis obtained comparing the list of differential expressed genes in met-SCC vs CT with the ones obtained from lymph-SCC vs CT and met-SCC vs lymph-SCC. As you can see form the histogram below (that we included in the manuscript as figure 3B), the 52% of significantly upregulated genes were in common between met-SCC vs CT and met-SCC vs lymph-SCC and demonstrated a further prompt of the angiogenetic pathway consistently with the worsening of the phenotype. At the same time, we observed that 48% of the downregulated genes in met-SCC vs CT were also significantly less expressed in lymph-SCC vs CT, indicating a progressive inhibition of the involved pathways as the disease worsens.

(iii) gene ontology network analysis: 

Gene ontology (GO) can give hints on the biological function of genes and proteins. These pathways can be named in an imprecise manner, such as some stated in the manuscript (“regulation of the immune response” or “regulation of cell proliferation”). In these cases, the effect of the gene set is not clear. In the discussion section, especially for the “regulation of the immune response” it seems as though this GO-term was interpreted as “antitumoral immune response”, while showing at the same time, that several genes herein are not active in “antitumoral immune responses”. All genes are individually discussed – here, it should always be included, in which patient group the respective gene was up- or downregulated.

We better clarified in which patient group genes were up- or down-regulated. We also disambiguated terminology whenever possible.

As the manuscript states, validated risk stratification strategies for cSCC are currently not available. Multiple studies have tried to meet this relevant subject, producing just as many different gene sets with the potential to predict the further course of cSCC disease – yet lacking clinical validation. Here, the authors announce a “molecular and histopathological signature” taking an adequate approach with data collection and transcriptomic analysis. The manuscript would benefit from distinctively stating the final combined (molecular and histopathological) signature and discussing the advantage of this signature in comparison to the multiple other signatures already published. In this context, it may be advisable to show the prognostic potential of the molecular and histopathological signature on their own and in combination. 

We acknowledge that the definition of “signature” is probably an overstatement, possibly confusing the readers, and we changed it into “characterization” both in the title and throughout the text. The current paper presents a pilot case-control study aimed at identifying genes and histopathological characteristics specific of metastatic SCCs. We plan to perform further studies based on consecutive case series aimed at validating the prognostic potential of such histological and molecular features.

Additional comments

In the results section (lines 117-119), supporting notes on how to fill out this section were accidentally included in the manuscript. 

Thank you, these lines have been removed.

Table 1 has a formatting / shifting error in the category “Breslow thickness”. In the following sections the met-SCC cohort is split into two groups – it may be advisable to show the clinical data for both subgroups separately. 

The formatting error has been corrected.

Interpretation of the Kaplan-Meier curves (lines 132-138) in the results section belongs in the discussion section. 

The interpretation of the curves has been moved to the discussion section.

Please state reasons for showing all 67 significantly deregulated genes in Table 2 and not only those 33 genes chosen according to the combination of p-value and fold change, which are included in the gene signature. 

We included in the table all 67 significantly deregulated genes since PCA analysis and gene ontology were performed on the entire list of DEGs. The 33 genes with the highest fold change represent a too low number of variables to perform relevant gene ontology analyses. In the previous version of the manuscript we tried to restrict the number of significant molecular features characteristic of met-SCC and we chose fold change as a parameter to do it. This has likely generated confusion in the Readers. In this revised version of the manuscript, we decided to describe the altered gene expression program that affects met-SCC in its complexity and we did not limit our results to genes with FC>2. Further studies will help to define selected molecular players driving cSCC features of aggressiveness.

In Figure 2, the reference group should be stated in the figure legend. Relevant genes mentioned in the discussion section may be labelled in the volcano plot. 

The required changes have been performed.

In Figure 3, labelling of the patient groups should be consistent with the rest of the manuscript. Additionally, the meaning of the ellipses should be explained in the figure legend. 

The required changes have been performed. We performed PCA plot with the function “prcomp” and we represented it using the package “ggbiplot” by R version 4.4.0. Clusters defined by ellipses were obtained by automatic classification of samples based on PCA scores. The center of each ellipse represents the barycentre of the points belonging to the same cluster, while the main axis represents the variance.

In Figures 4 and 5 all ladder plots should have the same size. Any significances should be added to the plots and stated in the figure legends (also if there are none). For better understanding, the title of the gene sets depicted in the ladder plots can be included in the figure legends. 

In order to clarify the figures and add significance to all comparisons, we replaced ladder plot with a boxplot for each gene.

Figure 5 is mirrored upside down. 

Figure 5 has been replaced.

For all figures, a higher resolution is necessary. 

All figures have been changed according to your comments.

In the results section (line 101/102), it is stated that the data are available at the GEO repository (release scheduled for Jan 2025). This should be included in the data availability statement. 

This information has been included in the data availability statement.

An ethics statement should be included concerning the informed consent of all patients or an ethical vote, that such a consent is not necessary due to the retrospective nature of the study.

This information has been included in the Informed Consent statement.

Reviewer 2 Report

Comments and Suggestions for Authors

The. manuscript is well-written. Clear results with good statistics. 

No other comments. 

Author Response

Thank you very much for your kind comments.

Reviewer 3 Report

Comments and Suggestions for Authors

Interesting retrospective study with good insights. Major limitation is the small sample size. Overall, the design is solid and balanced. Minor adjustments are needed to improve the draft. Here are some considerations: 

1. The study could benefit from a deeper exploration of the so called "high-risk features" outlined in the recent EADO-EADV guidelines (Stratigos 2023). Details such as the number of cases with tumor thickness exceeding 6 mm, desmoplasia, bone erosion, and known immunosuppression status would enrich the analysis. Were these data analysed at the time of histopath assessment? 

2. Please define depth of infiltration. Despite substantial overlap in most cases, in lesions exhibiting an exophytic growth pattern the vertical tumor thickness (measured as Breslow thickness) and depth of infiltration (determined by measuring the neoplastic mass below the granular layer of the adjacent uninvolved epidermis) can differ.

3. The authors state "No significant difference in terms of lympho-vascular and/or perineural invasion was detected". However, no details were reported in the table regarding the number of patients with these features.

4. How many cases of poorly differentiated SCC showing lymphvascular invasion were included? A recent study unveiled lymph-vascular invasion as the most important predictor of local and distante recurrence in G3-SCC. I would suggest a comment on this finding DOI: 10.1159/000535040

5. Were the risk factors for non complete excision assessed? For your reference please see DOI: 10.1111/jdv.18101

6. I would suggest to comment on the potential practical applications in clinical settings of GEP panles beyond mere profile description. Do the authors foresee a practical use of such profiles in clinical settings?

Comments on the Quality of English Language

Good

Author Response

Interesting retrospective study with good insights. Major limitation is the small sample size. Overall, the design is solid and balanced. Minor adjustments are needed to improve the draft. Here are some considerations: 

Thank you very much for your fruitful comments, which allowed us to improve manuscript quality.

  1. The study could benefit from a deeper exploration of the so called "high-risk features" outlined in the recent EADO-EADV guidelines (Stratigos 2023). Details such as the number of cases with tumor thickness exceeding 6 mm, desmoplasia, bone erosion, and known immunosuppression status would enrich the analysis. Were these data analysed at the time of histopath assessment? 

Several high-risk features described in the EADO-EADV guidelines have been further discussed. Lack of information regarding the listed characteristics has been listed as study limitation when not available. No cases of tumors with bone erosion were included in our cohort. As for the immune suppression status, due to the retrospective nature of the study, we only had availability of data on solid organ transplant recipients, with lacking data on other causes of immune suppression, such as -for example- hematological malignancies, HIV, chemotherapy, or immunosuppressive therapies for rheumatological disorders. Consequently, we decided not to proceed with statistical analysis on incomplete data.

  1. Please define depth of infiltration. Despite substantial overlap in most cases, in lesions exhibiting an exophytic growth pattern the vertical tumor thickness (measured as Breslow thickness) and depth of infiltration (determined by measuring the neoplastic mass below the granular layer of the adjacent uninvolved epidermis) can differ.

As indicated in Table 1, we assessed Breslow thickness. Despite we acknowledge that it can overestimate the depth of infiltration in exophytic masses, other measurements are not routinely performed nor were included in the study protocol.

  1. The authors state "No significant difference in terms of lympho-vascular and/or perineural invasion was detected". However, no details were reported in the table regarding the number of patients with these features.

We added more details on lymphovascular and perineural invasion: “Lymphatic or vascular involvement and perineural invasion were not shown in Table 1 because in both groups no patients presented with these histopathologic features”.

  1. How many cases of poorly differentiated SCC showing lymphvascular invasion were included? A recent study unveiled lymph-vascular invasion as the most important predictor of local and distante recurrence in G3-SCC. I would suggest a comment on this finding DOI: 10.1159/000535040

Thank you, we added a comment on such finding.

  1. Were the risk factors for non complete excision assessed? For your reference please see DOI: 10.1111/jdv.18101

We further discussed risk factors for non-complete excision.

  1. I would suggest to comment on the potential practical applications in clinical settings of GEP panles beyond mere profile description. Do the authors foresee a practical use of such profiles in clinical settings?

We better discussed the potential applications of gene expression profiles in the discussion and conclusions sections.

Round 2

Reviewer 1 Report

Comments and Suggestions for Authors

The authors decided to change the title / significance from "signature" to "characterization" of SCC features. This alteration is appropriate. 

Concerning figures: 

- Figures have been improved. Exchange of ladder blots with box plots is appreciated. 

- Caption of Fig 1 is missing. Unfortunately, I could not find the supplementary figures for elastosis and differentiation grade. 

- Fig 3: For the volcano plot, please include the reference group for the FC, otherwise the plot cannot be interpreted. 

- Fig 4: Please include the meaning of the ellipses in the figure legends of the PCA.

Other notes: 

- In line 186/7 there is an incomplete sentence. 

Comments on the Quality of English Language

Some minor English language errors remain, please check carefully. 

Author Response

Thank you very much for your prompt and constructive feedback. We performed some minor changes according to your kind suggestions in order to improve the quality of the manuscript. In particular:

- we included the complete scales for elastosis and differentiation in figure 1and added the correspondent figure legend

- we modified the volcano plot, in order to ease its interpretation. 

- we included the meaning of the ellipses in the figure legends of the PCA

- we checked all the text for minor spelling mistakes and/or incomplete sentences.